# Uncovering Plant Virus Species Forming Novel Provisional Taxonomic Units Related to the Family *Benyviridae*

**DOI:** 10.3390/v14122680

**Published:** 2022-11-29

**Authors:** Andrey G. Solovyev, Sergey Y. Morozov

**Affiliations:** 1A. N. Belozersky Institute of Physico-Chemical Biology, Moscow State University, 119992 Moscow, Russia; 2Department of Virology, Biological Faculty, Moscow State University, 119234 Moscow, Russia; 3All Russia Research Institute of Agricultural Biotechnology, 127550 Moscow, Russia

**Keywords:** RNA virus, benyviruses, movement gene modules, RNA polymerase, evolution, RNA helicase, phylogeny, land plants, algae

## Abstract

Based on analyses of recent open-source data, this paper describes novel horizons in the diversity and taxonomy of beny-like viruses infecting hosts of the plant kingdom (Plantae or Archaeplastida). First, our data expand the known host range of the family *Benyviridae* to include red algae. Second, our phylogenetic analysis suggests that the evolution of this virus family may have involved cross-kingdom host change events and gene recombination/exchanges between distant taxa. Third, the identification of gene blocks encoding known movement proteins in beny-like RNA viruses infecting non-vascular plants confirms other evidence that plant virus genomic RNAs may have acquired movement proteins simultaneously or even prior to the evolutionary emergence of the plant vascular system. Fourth, novel data on plant virus diversity highlight that molecular evolution gave rise to numerous provisional species of land-plant-infecting viruses, which encode no known potential movement genetic systems.

## 1. Introduction

*Benyvirus* is the only genus officially recognized by ICTV in the family *Benyviridae* (order *Hepelivirales*), with the species *Beet necrotic yellow vein virus* (BNYVV) being the type benyvirus [1,2]. The genus *Benyvirus* includes several viruses that have multipartite genomes and rod-shaped virions infecting plants and are usually transmitted by root-infecting vectors. Their single-stranded, positive-sense RNA genome has four segments containing a 5′-terminal “cap” structure and a 3′ poly(A) tract [1]. However, RNAs 1 and 2 may cause productive infection. RNA1 encodes a replicative protein with several typical domains, whereas RNA2 includes the 5′-terminal capsid protein (CP) ORF1 extended by a read-through ORF2, a Triple Gene Block (TGB) movement module and a cysteine-rich protein-encoding ORF6 (Figure 1) [1,3,4,5,6,7].

Our recent papers based on published metagenomic studies have revealed several novel plant virus RNAs encoding benyvirus-related replicative proteins and movement gene modules that are close to a TGB in structural, functional and phylogenetic aspects [8,9,10,11,12,13]. Metagenomic studies and phylogenetic analyses have also shown that many benyvirus-related sequences seem to correspond to novel non-classified phylogenetic *Benyviridae* clusters comprising viruses infecting a wide range of non-plant hosts, namely, insects and fungi [14,15,16,17,18]. In particular, such insect viruses include *Bemisia tabaci beny-like virus 6* (NCBI accession MW256699), *Bemisia tabaci beny-like virus 4* (MW256697), *Hubei Beny-like virus 1* (MK231108) and *Diabrotica undecimpunctata virus 2* (MN646771), and fungal viruses are represented by *Erysiphe necator associated beny-like virus 1* (MN617775), *Rhizoctonia solani beny-like virus 1* (MK507778) and *Sclerotium rolfsii beny-like virus 1* (MH766487) [15,19,20,21,22]. 

Sets of non-replicative proteins encoded by beny-like plant viruses, on the one hand, and insect/fungal viruses, on the other hand, are generally different from non-plant viruses containing no recognizable movement genes [15,20,21,22]. In this study, we comprehensively analyzed the gene organization of NCBI-annotated virus genomes and virus-like RNA assemblies (VLRAs), which were identified in transcriptome databases and found to encode replicative proteins related to those in viruses of the genus *Benyvirus*, and showed that the molecular evolution of these replicating RNAs gave rise to numerous provisional virus species, which could infect hosts of the kingdom Plantae and encode no TGB-like gene modules or other known movement genetic systems.

## 2. Materials and Methods 

Benyvirus-like nucleotide sequences were collected from the NCBI plant transcriptome database and 1KP transcriptome database (https://db.cngb.org/onekp/, accessed on 26 June 2022). The software used for the viral taxonomic classification is provided by NCBI (https://www.ncbi.nlm.nih.gov/taxonomy/?term=viruses/, accessed on 26 June 2022). Our searches were also extended to the most recent plant virus sequence database (https://riboviria.org/, accessed on 26 June 2022) [23]. Sequence comparisons were carried out using the BLAST algorithm (BLASTn and BLASTp) at the National Center for Biotechnology Information (NCBI). Open reading frames (ORFs) were identified using ORF Finder programs (http://www.bioinformatics.org/sms2/orf_find.html and https://www.ncbi.nlm.nih.gov/orffinder/, accessed on 26 July 2022). Conserved motif searches were conducted in CDD (http://www.ncbi.nlm.nih.gov/Structure/cdd/wrpsb.cgi, accessed on 26 July 2022) databases. Hydrophobic protein regions were predicted using Novoprolabs software (https://www.novoprolabs.com/tools/protein-hydropathy/, accessed on 26 July 2022). To assemble the 3′-untranslated regions (up to 100–150 nucleotides) of some nearly full-length plant VLRAs, SRA experiments linked to the TSA projects were downloaded using the fastq-dump tool of the NCBI SRA Toolkit (http://ncbi.github.io/sra-tools/, accessed on 26 July 2022). Read quality was checked with FastQC (https://www.bioinformatics.babraham.ac.uk/projects/fastqc/, accessed on 26 July 2022). De novo assembly of the 3′-untranslated regions of some VLRAs was carried out using SPAdes (http://cab.spbu.ru/software/spades/, accessed on 26 July 2022) [24] in “RNA mode”. Phylogenetic analysis was performed with “Phylogeny.fr” (a free, simple-to-use web service dedicated to the reconstruction and analysis of phylogenetic relationships between molecular sequences) by constructing maximum likelihood phylogenetic trees (http://www.phylogeny.fr/simple_phylogeny.cgi, accessed on 26 July 2022). Bootstrap percentages obtained with 1000 replications were used. Genome sequences of different benyviruses and beny-like viruses were downloaded from the GenBank database (see text below). 

The search for benyvirus-like RNA genomes was initially performed by NCBI TBLASTN analysis with the RNA-dependent RNA polymerase (RdRp) domain of the BNYVV replicative polypeptide as a query. This analysis across the breadth of Excavata (Discoba), Diaphoretickes (Cryptophyta, Rhizaria, Alveolata, Stramenopiles and Archaeplastida) and Amorphea (Fungi, Metazoa and Amoebozoa) [25,26] revealed hundreds of nucleotide contigs encoding proteins with more than 25% identity to the BNYVV RdRp domain (data not shown). RNA contigs under 500 nucleotides in length were excluded from analysis, and thus, the average sizes of the analyzed sequences were about 4500–5000 nucleotides. Further filtering based on pairwise protein sequence similarity NCBI BLASTp searches against the virus protein database (https://blast.ncbi.nlm.nih.gov/Blast.cgi?PAGE=Proteins&PROGRAM=blastp&BLAST_PROGRAMS=blastp&PAGE_TYPE=BlastSearch&BLAST_SPEC=blast2seq&DATABASE=n/a, accessed on 26 July 2022) for beny-like virus detection was necessarily performed to discriminate highly diverged Benyviridae-like viruses from the representatives of other virus families in the order *Hepelivirales* (*Hepeviridae*, *Matonaviridae*, etc.). For a description of the latter orders, see https://ictv.global/report/chapter/hepeviridae/hepeviridae and https://ictv.global/report/chapter/matonaviridae/matonaviridae (accessed on 26 July 2022).

The nucleotide sequences of contigs from the NCBI TSA database, which were first identified as VLRAs in the present work, can be found using the following accession numbers: red algae and plants—Agarophyton vermiculophyllum VLRA (GILD01050008 and GILD01008494, total 5120 nts), Lithophyllum crustose DN227823 VLRA (GHIV01061204, 5537 nts), Lithophyllum crustose DN218665 VLRA (GHIV01051861, 2575 nts), Gypsophila paniculata VLRA (GILV01257737, 5090 nts), Cenostigma pyramidale VLRA (GIYP01578425, 4555 nts), Salvia miltiorrhiza VLRA (GJJN01058736, 3972 nts), Rhyncholacis cf. penicillata Rhyc27837 VLRA (ICSC01056734, 8281 nts), Rhyncholacis cf. penicillata Rhyc16 VLRA (ICSC01000014, 7893 nts), Astragalus canadensis VLRA (GGNK01006219, 3313 nts), Melampyrum roseum VLRA (IADV01103213, 2174 nts), Striga hermonthica VLRA (ICPL01009187, 8031 nts), Coriandrum sativum VLRA (GGPN01001998, 8098 nts), Camellia reticulata VLRA (GEER01003429, 6431 nts), Viscum album VaGs28290 VLRA (GJLG01028288, 2781 nts), Daiswa yunnanensis VLRA (GFOY01013898, 2707 nts), Ophrys fusca VLRA (GHXI01129489, 7998 nts), Gymnadenia rhellicani VLRA (GHXH01324014, 7925 nts), Ophrys sphegodes VLRA (GHXJ01414654, 8005 nts) and Silene dioica VLRA (GFCG01071918, 9583 nts); arthropods—Xestocephalus desertorum VLRA (GELC01037548, 7329 nts), Pempsamacra sp. AD-2015 VLRA (GDNN01032725, 4921 nts), Eudarcia simulatricella VLRA (GEOF01029671, 2107 nts), Hydraena subimpressa VLRA (GDPN01031386, 5116 nts), Hypocaccus fitchi VLRA (GDLI01018707, 5112 nts), Porrostoma sp. AD-2015 VLRA (GDLM01020977, 5227 nts), Arrhenodes minutus VLRA (GDPV01017150, 5135 nts), Galerucella pusilla VLRA (HAMG01051171, 7316 nts) and Holacanthella duospinosa VLRA (GFPE01052446 7680 nts).

To generate a phylogenetic tree, we aligned some of the revealed RdRp domains translated from the large number of found Archaeplastida VLRAs and selected VLRAs of other origins with selected known virus RdRp protein sequences taken from NCBI/GenBank. In total, we included 97 RdRp-encoding contigs (including the outgroup species *Beet yellows closterovirus*) that represent annotated and yet-uncharacterized virus species. VLRA transcripts and annotated virus genomes presented in the tree were taken from 44 Archaeplastida species, while additional included RdRp regions were from viruses with non-plant hosts (mainly fungi and arthropods) and VLRAs with unknown hosts (taken from https://riboviria.org/ (accessed on 26 July 2022)). Among Archaeplastida sequences, we included three sequences from red algae, four from Chara algae and five from non-flowering land plants. The resulting tree is presented in Figure 2. 

## 3. Results 

The two most diverged benyvirus-like VLRAs were revealed in marine organisms. One of these was from the Rhodophyta-specific VLRA of the marine algae *Agarophyton vermiculophyllum* (Figure 2). Another one represented by the VLRA sequence of the marine euglenozoan protist *Diplonema papillatum* also formed no clusters with other sequences. This species belongs to Discoba, a lineage hypothetically proximal to the eukaryotic root, while red algae (Rhodophyta) are the most ancient marine organisms in the kingdom Plantae (Archaeplastida) (https://www.algaebase.org/browse/taxonomy/, accessed on 26 July 2022) [26,27]. Therefore, these observations indicate the emergence of beny-like virus replicases before the diversification of eukaryotes or at least the kingdom Plantae. The third basal branch of the tree is again formed by a single representative, the moss *Schwetschkeopsis_fabronia* VLRA (Figure 2). Very recently, this VLRA from the 1KP transcriptome project was annotated as the species *Leucodon julaceus associated beny-like virus* (LjBV) [16]. Other well-separated tree branches include more than one sequence representative. 

Taking plant-specific virus and VLRA sequences as references, we could discriminate several mixed and well-separated tree clusters: (i) the beny-like RdRp encoded by the VLRA from the non-vascular plant *Pellia neesiana* representing the class Jungermanniopsida is included in the mixed branch together with many fungal viruses and some arthropod-specific viruses; (ii) the Pistacia ribo-like virus RdRp forms a common cluster with many fungal and arthropod-specific viruses; (iii) the flowering plant *Cenostigma pyramidale* VLRA RdRp is included in a common cluster with some fungal viruses; (iv) the flowering plant *Salvia miltiorrhiza* VLRA RdRp forms a common cluster with some fungal viruses and red algae VLRAs (Figure 2). Because the phylogenetic placement of the above-mentioned plant viruses is rather close to viruses infecting very distant hosts, from red algae to arthropods, there is a possibility that plant transcriptomes have been contaminated by RNA molecules from other non-plant species as a result of improper sample collection, symbiosis, parasitism or other factors. However, recent estimations of potential contaminations in the 1KP plant transcriptome database suggest that host cross-species transmission in plant and non-plant viruses prevails over different cases of transcriptome artifact contamination [16].

The largest cluster in the presented phylogenetic tree is composed of RdRp sequences of Viridiplantae viruses only. It contains two main sub-branches including Chara-virus-related sequences and land plant virus sequences (Figure 2). The land plant sub-branch includes RdRp sequences of viruses from the genus *Benyvirus* and some non-classified viruses from *Benyviridae*. Moreover, it contains VLRA-derived sequences that may represent virus species forming novel putative taxa in the family *Benyviridae* (see below). 

### 3.1. Species Forming the Most Basal Branch of the Phylogenetic Tree

The sequence forming the most basal branch of the tree is from the Rhodophyta transcriptome of the marine algae *A. vermiculophyllum* (NCBI accessions GILD01050008 and GILD01008494) (Figure 2). This VLRA contains four ORFs (Figure 3). The incomplete ORF1 includes three conserved domains: a viral helicase 1 domain (HEL, pfam01443, E-value 1.09e-20), FtsJ-like methyltransferase (AdoMet methyltransferase superfamily) (FtsJ, pfam01728, E-value 1.69e-12) and an RdRp core motif (pfam00978, E-value 4.33e-09), but it lacks a viral methyltransferase domain (MTR, pfam01660), probably because of sequence truncation (Figure 3). 

Pairwise sequence comparisons (Appendix A) show that the Rhodophyta-specific viral HEL SF-1 domain is most similar to benyvirus-like species from plants and somewhat less similar to another distantly related Rhodophyta-specific VLRA (see below) as well as arthropod-specific viruses. In general, these data confirm the rather distant relation of the A. vermiculophyllum VLRA to the rest of the beny-like viruses with different hosts. Indeed, pairwise sequence comparisons show that the BNYVV core RdRp domain has 28% identity to the type members of families *Hepeviridae* and *Matonaviridae* (species *Human hepatitis E virus* and *Rubella virus*), whereas A. vermiculophyllum VLRA RdRp has 33–36% identity to BNYVV and other approved members of the genus *Benyvirus*. It can be hypothesized that this potential Rhodophyta-specific virus can be regarded as a founding member of a new potential taxon (for example, a subfamily of *Benyviridae*) in the order *Hepelivirales.*

The occurrence of the FtsJ-like methyltransferase domain in the A. vermiculophyllum VLRA marks its difference in the replicase domain arrangement from benyviruses, which encode no such protein domain [1]. However, this domain has been revealed, particularly in replicase proteins of some members of the family *Kitaviridae* (order *Martellivirales*) [28] and related negeviruses [29]. Amazingly, although FtsJ of the A. vermiculophyllum VLRA has only marginal amino acid sequence similarity to this domain in *Martellivirales*, it shows significant similarity to insect DNA-virus-encoded FtsJ proteins, particularly those of *Choristoneura rosaceana nucleopolyhedrovirus* (genus *Alphabaculovirus*; family *Baculoviridae*) (YP_008378434, 40% identity, E-value 2e-07), Anticarsia gemmatalis nucleopolyhedrovirus (YP_803463, 37% identity, E-value 2e-06) and *Neophasia sp. alphabaculovirus* (QBC76063, 38% identity, E-value 3e-06). It should be noted that although the RdRp domain of the Rhodophyta-specific VLRA shows obvious similarity to benyvirus replicases, its replicase seems to contain no papain-like proteinase domain, which is conserved in benyvirus replicases [1] (Figure 1). 

ORF2 of the A. vermiculophyllum VLRA encodes a capsid-like protein (CPL) (Figure 3) of the tobamo-like group that is rather common for rod-shaped viruses [30]. Although the CPL of this VLRA (225 aa in length) has only marginal amino acid sequence identity to CPs of the genus *Benyvirus*, it shows significant similarity to tobamovirus CPs, particularly the cucumber fruit mottle mosaic virus CP (genus *Tobamovirus*; family *Virgaviridae*) (QSM07163, 29% identity, E-value 4e-06), and even higher identity to insect beny-like viruses, particularly the Guiyang benyvirus 1 CPL (family *Benyviridae*) (UHK03085, 40% identity, E-value 7e-21). However, additional TBLAST analysis of the NCBI TSA database shows that the most similar relatives of the Agarophyton vermiculophyllum VLRA CPL are encoded by truncated VLRAs of another red algal species, *Rhodonematella subimmersa* (GFTI01074942, 44% identity, E-value 3e-12), and the flowering plant *Cenostigma pyramidale* (GIYP01218754, 48% identity, E-value 1e-11). The predicted ORF3 in the A. vermiculophyllum VLRA overlaps ORF2 (Figure 3) and encodes a positively charged protein of 135 residues with pI = 11.37, having an uncharged N-terminal region (Figure 4). ORF4 codes for a small protein of 128 residues with no characterized domains. 

Another basal-branch-forming VLRA sequence found for the marine euglenozoan protist *D. papillatum* (NCBI accession GJNJ01037323) contains four ORFs (Figure 3). The incomplete ORF1 contains the conserved RdRp domain with a GDD signature. The short ORF2 encodes a small protein of 47 residues containing no characteristic protein signatures. ORF3 overlaps ORF2 (Figure 3) and codes for a positively charged protein of 63 residues with pI = 11.73, having a nuclear localization signal (NLS according to NLStradamus software) (Figure 4). These properties allow us to speculate that the ORF3 protein is involved in silencing suppression, as has been shown for some plant viruses [31,32]. ORF4, overlapping with ORF3 (Figure 3), encodes a small protein (44 amino acids in length) with a C-terminal hydrophobic segment (Appendix A). It should be noted that the beny-like D. papillatum VLRA may represent the second example of a positive-stranded RNA virus infecting Discoba, a lineage hypothetically proximal to the eukaryotic root [24]. To date, only members of the family *Narnaviridae* have been revealed in these hosts [33]. As we found no ORFs coding for capsid-like proteins in the Diplonema papillatum VLRA, it can be hypothesized that this potential virus may exist as a “naked” virus, such as narnaviruses [34]. Additionally, assuming a rather low identity level between core RdRp domains of the D. papillatum VLRA and BNYVV (27%), it can be speculated that this potential euglenozoan protist virus may represent the founder of a new family in order *Hepelivirales*.

### 3.2. Cluster of the Phylogenetic Tree including Rhodophyta-Specific VLRAs

A cluster of nine RdRp sequences includes two additional Rhodophyta-specific VLRAs found for Lithophyllum crustose, as well as sequences of fungi-specific viruses and a higher-plant-specific VLRA from *Salvia miltiorrhiza* (family *Lamiaceae*) (Figure 2). Among two Rhodophyta-specific VLRAs, the L. crustose DN227823 VLRA (GHIV01061204) is longer (likely close in length to the full-size genome) and includes 5537 bases. This VLRA encodes two proteins. The incomplete ORF1 includes two conserved domains: a viral helicase 1 domain (HEL, pfam01443, E-value 5.81e-18) and an RdRp core motif (pfam00978, E-value 7.85e-12) (Figure 3). Unlike the Agarophyton VLRA (see above), this replicase encodes no FtsJ-like methyltransferase domain. Additionally, in contrast to the Agarophyton VLRA HEL domain, the Lithophyllum VLRA HEL domain is closer to replicases of arthropod-specific viruses than to those of plant-specific ones (Appendix A). ORF2 of the Lithophyllum crustose DN227823 VLRA encodes a negatively charged protein (pI 5.6; 165 aa in length) with no detectable sequence similarity to known virus proteins. 

A land-plant-infecting virus in this cluster is represented by the Salvia miltiorrhiza VLRA, which is closer, according to the RdRp sequence, to fungal viruses than to Rhodophyta-specific VLRAs (Figure 2). An RdRp domain with a GDD signature is found in the C-terminal portion of the incomplete ORF1 (Figure 3). The short ORF2 encodes a small positively charged protein (pI = 10.4) of 74 residues containing no characteristic protein signatures. ORF3 codes for a highly hydrophobic, potentially membrane-binding protein of 118 residues in length (Appendix A). This ORF is almost completely overlapped by ORF4 (Figure 3), which encodes a positively charged protein (pI = 10.7; 307 aa in length). This protein contains an NLS sequence motif (RGRGRGRGGFRGRGR, score 0.84) overlapping with the typical N-terminal RGG/RG motif, which is known to participate in protein–RNA interactions of many eukaryotic cell proteins [35,36]. Interestingly, RGG/RG motifs were found to take part in genomic RNA binding by the betacoronavirus nucleocapsid protein [37]. 

Assuming the percentage of identity between the RdRp core domain of BNYVV and those of the Lithophyllum crustose DN227823 VLRA (41%) and the Salvia miltiorrhiza VLRA (39%), it can be speculated that the cluster described above may contain viruses of novel subfamilies in the family *Benyviridae*. These considerations are indirectly supported by the fact that some NCBI-approved unclassified members of this family (species *Entomophthora benyvirus E* isolate Cho, *Leucodon julaceus beny-like virus***,**
*Bemisia tabaci beny-like virus 4*, *Wallace’s spikemoss associated beny-like virus*, *Bemisia tabaci beny-like virus 4* and *Rhizoctonia solani beny-like virus 2*) (https://www.ncbi.nlm.nih.gov/Taxonomy/Browser/wwwtax.cgi?id=1513233 (accessed on 26 July 2022)) express similar or somewhat higher RdRp sequence identities to BNYVV (31–46%).

### 3.3. Large Cluster of RdRp Domains Encoded by Viridiplantae-Specific Viruses and VLRAs

The largest cluster of the phylogenetic tree includes members of the genus *Benyvirus* along with numerous other plant-infecting viruses (Figure 2). Two viruses exhibiting high sequence identity and similarity in genome organization to *Benyvirus* members are represented by the bipartite species *Wheat stripe mosaic virus* [38,39] and the fern Asplenium nidus VLRA [11]. The latter VLRA has been recently annotated as the species *Fern benyvirus* belonging to the family *Benyviridae* [16]. In this cluster, the viruses most distantly related to *Benyvirus* genus members are closely positioned to the monopartite species *Chara australis virus*, which can be tentatively attributed to the Charavirid group (see below). Generally, in this large cluster, we provisionally named three additional groups of putatively monopartite and bipartite plant viruses: Tecimovirids (Tetra-cistron movement block-containing viruses), Binamovirids (Binary movement block-containing viruses) and Reclovirids (Red clover RNA virus 1) (Figure 5A). 

#### 3.3.1. Charavirids 

*Chara australis virus* (CAV) [40] has an RNA genome of around 9000 nucleotides (accession JF824737) and encodes a large replicative protein, which shows a relationship with RNA polymerases of benyviruses. Charophyte algae are considered the ancestors of land plants, and Chara viruses may be evolutionarily related to ancestor virus species that infected the first plants colonizing terrestrial habitats [40]. Interestingly, beny-like Chara viruses are distributed around the globe, since close species were found in Australia and North America (*Charavirus-Canada*) [41]. 

Previously, we revealed closely related viral RNA metagenomic sequences in the NCBI Sequence Read Archive (SRX8007769), BioProject accession PRJNA615325 (*Charavirus-Tibet*) (Figure 2 and Figure 5A). These data have been derived from samples of fish gills (*Gymnocypris namensis*) collected in Qinghai Lake in the Tibet Autonomous Region of China. A very close virus isolated from the fish body has been recently annotated as *Charavirus-Namtso* (MW483685). The largest encoded protein of these monopartite viruses shows a relationship with RNA polymerases of benyviruses (Figure 2), while the capsid protein is distantly related to the tobamovirus CP [40,41]. Two additional open reading frames (ORFs) code for an RNA helicase and a protein of unknown function. Importantly, this non-replicative “accessory” helicase is related to helicases of the SF-2 superfamily, in contrast to “accessory” TGB1 helicases belonging to SF-1 (Figure 5B) [3,41]. Apparently, the replicase and the “accessory” helicase belong to different lineages of positive-stranded viruses in the orders *Hepelivirales* and *Amarillovirales*, respectively, whereas the “tobamo-like” CPL probably emerged in a common ancestor of the *Hepelivirales* and *Martellivirales* orders [23,41]. It should be noted that VLRAs of unknown origin (ND 427881, ND 201999 and ND 356784 (https://ribiviria.org (accessed on 26 July 2022)), see Figure 2) have a genome organization resembling that of *Chara australis virus.* The Charavirid RdRp core domain has 34–36% identity to that of BNYVV, and this cluster can be speculated to include viruses of a new genus/subfamily in the family *Benyviridae.*

#### 3.3.2. Goji Berry Chlorosis Virus

*Goji berry chlorosis virus* (GBCV) is positioned rather close to members of the genus *Benyvirus* in the phylogenetic tree (Figure 2 and Figure 5A). The GBCV RNA genome encodes six polypeptides [42]. The GBCV replicase (ORF1 protein) includes three conserved domains of RNA polymerases, namely, a viral helicase 1 domain (HEL, pfam01443, E-value 1.01e-25) and the RdRp core motif (pfam00978, E-value 1.07e-10), but lacks a viral methyltransferase domain (MTR, pfam01660) (Figure 5B). This domain organization is quite similar to the replicase of the beny-like Rhodophyta-specific VLRA of the marine algae *A. vermiculophyllum* (see above). The GBCV RdRp core domain has 52% identity to that of BNYVV. The GBCV CP (ORF2 protein) represents the “tobamo-like” CPL [42] with the closest sequence similarity to CPLs of arthropod-infecting members of the order *Martellivirales*, particularly the *Abisko virus* CPL (YP_009408587, 29% identity, E-value 2e-06), *Xiangshan martelli-like virus 2* CPL (UDL14010, 28% identity, E-value 1e-04) and *Bemisia tabaci virga-like virus 2* CPL (QWC36454, 25% identity, E-value 1e-06). Strikingly, the CPL-encoding gene in the GBCV genome is duplicated, and its second copy is represented by ORF3, a protein product of which is most closely related to the *Tobacco rattle virus* CP (genus *Tobravirus*) (AAC02063, 28% identity, E-value 3e-04). The ORF4 and ORF5 proteins have no significant similarity to known virus proteins and are marginally related to plant myosin-11 (ORF4 protein) and plant kinesin calmodulin-binding protein (ORF5 protein) (data not shown). ORF6 has been experimentally shown to encode a silencing suppressor protein [42]. This genome organization suggests that GBCV may represent a recombinant between viruses from the families *Benyviridae* and *Virgaviridae* (more specifically, the order *Martellivirales*) [42]. 

#### 3.3.3. Tecimovirids

Recently, we predicted a novel putative movement gene module, called the tetra-cistron movement block (TCMB), in the VLRAs of the moss *Dicranum scoparium* and the Antarctic flowering plant *Colobanthus quitensis* [8]. These putative bipartite viruses contain RNA components with a gene organization related to genomes of the genus *Benyvirus*. Replicases of tecimovirids contain RdRp domains with an obviously close relationship to those of viruses in the genus *Benyvirus* (Figure 2 and Figure 5). In particular, the RdRp core domain has 63% identity to that of BNYVV. Like RNA2 of the genus *Benyvirus*, tecimovirid-encoding RNA2 has a 5′-terminal CPL gene, and its RNA helicase gene is included in the 3′-terminal region and is followed by two small overlapping cistrons encoding hydrophobic proteins with distant sequence similarity to the TGB2 and TGB3 proteins. However, unlike the TGB, the TCMB also includes a fourth 5′-terminal gene preceding the helicase gene and coding for a protein showing similarity to the double-stranded RNA-binding proteins of the DSRM AtDRB-like superfamily [8].

#### 3.3.4. Binamovirids

This group of monopartite capsidless viruses is composed of three plant VLRAs (Figure 2 and Figure 5), which were initially revealed upon TBLASTN searches of NCBI and 1KP databases with *Hibiscus green spot virus* (HGSV) “accessory” SF-1 helicase as a query [9]. HGSV belongs to the family *Kitaviridae* (order *Martellivirales*) [28]. The HGSV RNA2-encoded “accessory” helicase is the first gene of the binary movement block [12,13] and shows the most pronounced similarity to three long VLRAs of the plants *Lathyrus sativus* (NCBI accession GBSS01016353, 7970 nucleotides), *Quercus castanea* (NCBI accession GHJU01198988, 7776 nucleotides) and *Litchi chinensis* (1KP accession WAXR-2010981, 7388 nucleotides). The RdRp core domain of these viruses has 50–52% identity to that of BNYVV. The BMB helicase sequence is somewhat closer to the tecimovirid “accessory” helicase than to the TGB1 helicase [8]. The replicase of binamovirids includes three conserved domains: a viral methyltransferase domain (MTR, pfam01660), a viral helicase 1 domain (HEL, pfam01443) and an RdRp core motif (pfam00978) (see Figure 1 from [9]). 

Recently, an additional plant VLRA encoding a binary movement block was revealed in the transcriptome of *Shorea curtisii* (accession GJMJ01032282). The unusual organization of this VLRA showed plus and minus chains of the RNA genome assembled in a long hairpin (data not shown). However, despite this probable artifact of sequencing data assembly, the VLRA code for ORF1 replicase with high similarity to the binamovirids *Lathyrus sativus*, *Quercus castanea* and *Litchi chinensis* (RdRp core domain identity 68–75%) and ORF2/3 proteins, which are similar to the BMB1 (Appendix A) and BMB2 proteins, respectively.

Despite the fact that binamovirids are quite close to members of the genus *Benyvirus* and tecimovirids, their genomes encode no coat protein. To check the possibility of the presence of additional RNA components in the genomes of binamovirids, we performed a BLAST analysis of transcriptome databases using the 3′-terminal regions of their replicase-encoding RNA segments as queries. However, this analysis revealed no additional genomic segments. Instead, this search showed the nucleotide sequence conservation of the extreme 3′-terminal areas between binamovirids, which confirms their close evolutionary relatedness (Figure 6). 

#### 3.3.5. Reclovirids

The largest subcluster of the phylogenetic tree showing benyvirus and beny-like RdRps is represented by numerous plant-specific VLRAs along with three annotated virus sequences, namely, *Red clover RNA virus 1* (MG596242), *Arceuthobium sichuanense virus 3* (BK059270 [43]) and *Dactylorhiza hatagirea beny-like virus* (BK013327) [44]) (Figure 2 and Figure 5). Strikingly, none of these 22 viruses code for genes of the CP or potential movement proteins. Therefore, we checked the possibility of the presence of additional RNA components in the genomes of reclovirids. As in the case of binamovirids, BLAST analyses of transcriptome databases using the 3′-terminal regions of their replicase-encoding RNA segments as queries have revealed no additional genomic segments. Strikingly, our analyses have demonstrated the nucleotide sequence conservation of the extreme 3′-terminal areas between a dozen reclovirids (Figure 7). Moreover, virus genomes with conserved 3′-non-coding regions are scattered throughout almost the whole reclovirid cluster (Figure 8).

To delve deeper into the monopartite genomic organization of previously poorly characterized reclovirids, we initially compared ORF maps within three groups including annotated virus sequences and the most closely related (according to the phylogenetic tree) VLRAs, namely, *Red clover RNA virus 1* with *Rhyncholacis penicillata* Rhyc27837 VLRA (ICSC01056734) and *Rhyncholacis penicillata* Rhyc16 VLRA (ICSC01000014); *Dactylorhiza hatagirea beny-like virus* with *Gymnadenia rhellicani* VLRA (GHXH01324014), *Ophrys fusca* VLRA (GHXI01129489) and *Ophrys sphegodes* VLRA (GHXJ01414654); and *Arceuthobium sichuanense virus 3* with *Viscum album* VLRA (GJLG01028288), *Atriplex prostrata* VLRA (EPVF-2046303), *Leontopodium alpinum* VLRA (DOVJ-2063722), *Silene dioica* VLRA (GFCG01071918) and *Astragalus canadensis* VLRA (GGNK01006218) (Figure 2 and Figure 8).

*Red clover RNA virus 1* RNA (7190 bases in length) encodes three proteins (Figure 3). The 5′-terminal ORF1 codes for RNA polymerase (2178 aa) with two characteristic domains: a viral helicase 1 domain (HEL, pfam01443, E-value 1.13e-18) and an RdRp core motif (pfam00978, E-value 4.10e-17). ORF2 and its nested ORF3 encode two hydrophobic proteins of 112 and 66 residues, respectively (Figure 3). We presume that both proteins represent “orphan” hydrophobic viral proteins [11] and have membrane-binding properties. 

The *Rhyncholacis penicillata* Rhyc27837 VLRA (8274 bases in length), despite the close amino acid sequence similarity and domain organization of the encoded RNA polymerase (ORF1 protein) to *Red clover RNA virus 1* RNA polymerase (identity 50% for the full-length proteins), codes for two proteins only (Figure 3). The ORF2 protein of the *R. penicillata* Rhyc27837 VLRA (313 residues in length) exhibits no similarity to the *Red clover RNA virus 1* ORF2 and 3 proteins. The *R. penicillata* Rhyc16 VLRA (7871 bases in length) also encodes two proteins, and its ORF1-encoded RNA polymerase has 58% identity of the full-length protein to the *R. penicillata* Rhyc27837 ORF1 protein. The ORF2 protein has no sequence similarity to *R. penicillata* Rhyc27837 VLRA-encoded ORF2; nevertheless, both proteins possess potential membrane-bound segments in the N- and C-terminal parts (Appendix A).

*Dactylorhiza hatagirea beny-like virus* RNA (7749 bases in length), which represents an isolated group of reclovirids (Figure 2 and Figure 5), encodes two proteins. ORF1-encoded RNA polymerase (2283 aa) has two characteristic domains, a viral helicase 1 domain (HEL, pfam01443, E-value 3.06e-18) and an RdRp core motif (pfam00978, E-value 6.81e-13). The ORF2-encoded protein is predicted to have hydrophobic regions (137aa) (Appendix A). ORF1 proteins of the *Gymnadenia rhellicani* VLRA, *Ophrys fusca* VLRA and *Ophrys sphegodes* VLRA belonging to the same group of reclovirids (Figure 5) have similar domain organizations (Figure 5A). The three above-mentioned reclovirid viruses show lower sequence identity (50–53%) in pairwise sequence comparisons with the BNYVV RdRp core domain in comparison with viruses belonging to the genus *Benyvirus* (73–95%) and can tentatively be regarded as members of another genus of *Benyviridae*.

Interestingly, we have found that the N-terminal 60% of the ORF2 protein (excluding its highly hydrophobic C-terminal segment (Appendix A)) shows obvious sequence similarity to the ORF2 proteins encoded by VLRAs identified for orchids *Gymnadenia rhellicani*, *Ophrys fusca* and *Ophrys sphegodes*. Moreover, similar proteins are encoded by three additional truncated VLRAs lacking the RdRp-encoding region. All of these proteins contain the characteristic CX(3)CX(10)CX(3)C motif in the N-terminal region (Figure 9) and hydrophobic segments in the C-terminal half (Appendix A). These cysteine-rich motifs resemble Cys-His fingers, which may be involved in nucleic acid binding and protein–protein interactions [45]. 

*Arceuthobium sichuanense virus 3* RNA (7341 bases in length) (Figure 2 and Figure 5) encodes two proteins: ORF1-encoded RNA polymerase (2071 aa) with two typical domains, a viral helicase domain (HEL, pfam01443, E-value 3.06e-18) and an RdRp core motif (pfam00978, E-value 6.81e-13), and an ORF2-encoded hydrophobic protein (137aa). Among related subcluster groups (Figure 2 and Figure 5), the *Arceuthobium sichuanense virus 3* ORF2 protein has the only counterpart with obvious similarity (identity 38%) found in the *Viscum album* VLRA (hydrophobic ORF2 protein) (Appendix A). Four other viruses close to *Arceuthobium sichuanense virus 3* encode either one ORF2 hydrophobic protein, similar to the *Leontopodium alpinum* VLRA (DOVJ-2063722), or two ORF2 and ORF3 hydrophobic proteins, with no sequence similarity to the *Arceuthobium sichuanense virus 3* ORF2 protein, as in the case of the *Atriplex prostrata* VLRA (data not shown).

Finally, it should be noted that among other Reclovirids included in Figure 2 and Figure 5, a rather common genome organization toward the 3′-terminus from ORF1 (the RNA polymerase gene) is revealed. All of these representatives encode one or two orphaned proteins usually including potential membrane-bound hydrophobic segments (data not shown).

## 4. Discussion

The data presented in this paper expand the known host range of the multipartite viruses in the family *Benyviridae* to include algae hosts. First, the phylogenetic analysis confirms the placement of monopartite charophyte algae-infecting viruses as a separate branch in the phylogenetic tree of beny-like viruses. These algae viruses are tentatively named “Charavirids”. Second, the presented analyses indicate that benyvirus-like replicases started their evolutionary radiation in red algae (Rhodophyta), which are the most ancient organisms in the kingdom Plantae (Archaeplastida) (https://www.algaebase.org/browse/taxonomy/, accessed on 26 July 2022). Rhodophyta-infecting potentially monopartite beny-like viruses are revealed in two clusters of the phylogenetic tree based on the RdRp core domain. We have found two red algae species carrying potential beny-like viruses, namely, *Lithophyllum crustose* and *Agarophyton vermiculophyllum*. The latter virus is positioned in the most basal part of the RdRp phylogenetic tree, and the corresponding VLRA also encodes a tobamo-like CPL, as is known for several members of the genus *Benyvirus* and some beny-like plant and insect viruses. In particular, the tobamo-like CPL is also encoded by the genomes of Charavirids (see above). Generally, the observed wide distribution of tobamo-like CPL genes [30] in monopartite and multipartite viruses of the family *Benyviridae* (order *Hepelivirales*) and family *Virgaviridae* (order *Martellivirales*) strongly suggests that horizontal gene transfer has played a significant role in the evolution of these virus taxa [15]. The divergence of the Charavirid CP from that of tobamoviruses is estimated to have occurred 212 million years ago (mya) [41]. On the other hand, the time of divergence between charavirus/benyvirus RdRp (order *Hepelivirales*) and virga-like RdRp genes (order *Martellivirales*) is estimated to be ~900 mya [41]. Therefore, benyvirus-like replicating RNAs likely started their evolutionary history in the late Precambrian, i.e., 850–1100 mya [27]. It should be noted that the evolutionary emergence of marine red algae, ranging from single-celled species to multicellular “plant-like” organisms, is estimated to have occurred approximately 1000–1600 mya [27,46,47]. Therefore, the divergence of replication proteins in viruses of the orders *Hepelivirales* and *Martellivirales* could have occurred in marine red algae species. In support of the proposed role of Rhodophyta as a host for common ancestors of *Hepelivirales* and *Martellivirales*, both types of viruses are found in present-day red algae hosts [48]. 

The analysis presented in this paper also highlights important clues to the evolution of plant cell-to-cell movement systems. First, the available data suggest that MP evolution started at least in lower land plants, as in these hosts, we have found typical movement gene blocks, such as the TCMB in Bryophyta (VLRA of the moss *Dicranum scoparium*) and the TGB in Polypodiopsida (*Fern benyvirus*) [8,11].

Novel data on plant virus diversity suggest that molecular evolution may have given rise to numerous potential plant-infecting virus species, which encode no known movement genetic systems. Previously, this fact has been a point of discussion. In fact, it is known that persistent plant viruses lack cell-to-cell movement systems and do not cause visible symptoms; accordingly, they are transmitted only vertically via gametes. Persistent plant viruses have been found in a few virus families, such as *Endornaviridae and Partitiviridae* [49,50,51,52]. Further, three large flavi-like replicating RNAs have been isolated from symptomatic plants; these RNAs, encoding single polymerase ORFs and no detectable CP and MP genes, have been termed *Snake River alfalfa virus*, *Carrot flavi-like virus 1* and *Gentian Kobu-sho-associated virus* [53,54,55,56]. It is proposed that these viruses could be arthropod-transmitted and spread inside the plant body with the help of other coinfecting viruses.

Similar to many persistent plant viruses, all Reclovirid replicating RNAs have been identified by high-throughput sequencing [43,44]. Therefore, these RNAs can be envisaged as spreading throughout plant bodies with the help of other viruses, as proposed for known persistent viruses. On the other hand, potentially monopartite Reclovirids can be proposed to use novel, yet-undescribed movement protein systems. Indeed, their polymerases are clearly related to benyviruses and beny-like viruses exploiting movement gene blocks, TGB, BMB or TCMB, which encode one, two or three small proteins with hydrophobic membrane-bound segments. As we have recently demonstrated, such proteins may account for the induction of ER tubule constrictions and increase the PD permeability, thereby facilitating intercellular viral spread, as has been found for the TGB2 and BMB2 proteins [13,57]. Thus, we hypothesize that similar functions can be performed by a broad range of Reclovirid hydrophobic orphan proteins [10]. Apparently, further detailed experimental studies are required to verify whether these proteins can serve as virus MPs.

## 5. Conclusions

In this work, we compared the genome sequences (mostly partial) of some novel plus-RNA viruses from red algae, a marine euglenozoan protist and land plants. Despite the concerns related to the potential risks of contig misassembly and contamination and the existence of unknown genome segments, in general, the reported sequence analyses of beny-like genomes confirm previously reported evolutionary relationships between viruses of non-flowering plants, flowering plants, arthropods and fungi, as well as suggest novel evolutionary links between viruses of red algae and land plants. We proposed several viruses as founding members of new potential taxa (subfamilies or genera) in the virus family Benyviridae. Additionally, the identification of gene blocks coding for movement proteins in viruses of non-vascular plants indicates that beny-like viruses may have acquired movement proteins prior to the emergence of the plant vascular system.

## Figures and Tables

**Figure 1 viruses-14-02680-f001:**
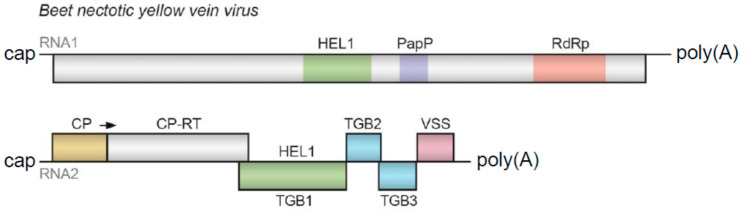
Gene organization of principal RNA genomic components of *Beet necrotic yellow vein virus*. Genes are shown as boxes with the names of the encoded proteins. Domains of RNA replicase ORF encoded by RNA1 include helicase superfamily 1 (HEL1), papain protease (PapP) and RNA-dependent RNA polymerase core domain (RdRp). Genes of proteins potentially involved in cell-to-cell movement (TGB) are shown in green and blue. VSS—virus silencing suppressor; CP-RT—capsid protein read-through protein.

**Figure 2 viruses-14-02680-f002:**
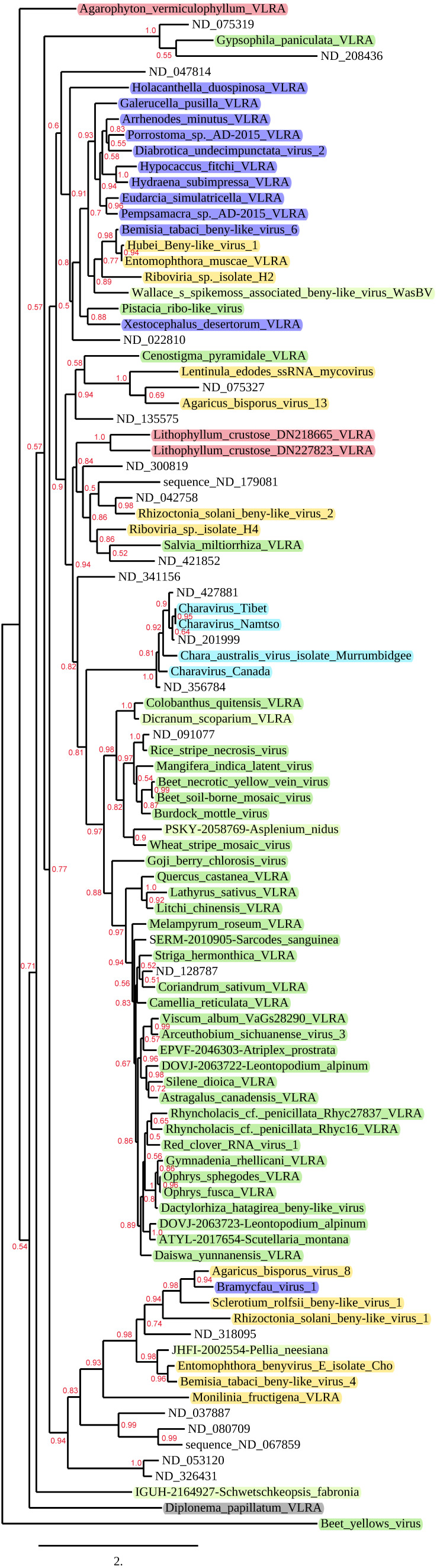
Phylogenetic analysis of the RdRp domains derived from the aligned deduced amino acid sequences of some benyviruses, benyvirus-like VLRAs and the selected beny-like viruses. The rooted phylogenetic tree (with the closterovirus species *Beet yellows virus* as the outgroup) was constructed using the maximum likelihood method based on amino acid sequence alignments. The bootstrap values obtained with 1000 replicates are indicated on the branches, and branch lengths correspond to the branch lines’ genetic distances. The genetic distance is shown by the scale bar at the lower left. Host specificity of the selected viruses is indicated by colors: green—flowering plants; light green—non-flowering plants; light blue—charophyte algae; pink—red algae; blue—arthropods; yellow—fungi; shading—euglenozoan protist.

**Figure 3 viruses-14-02680-f003:**
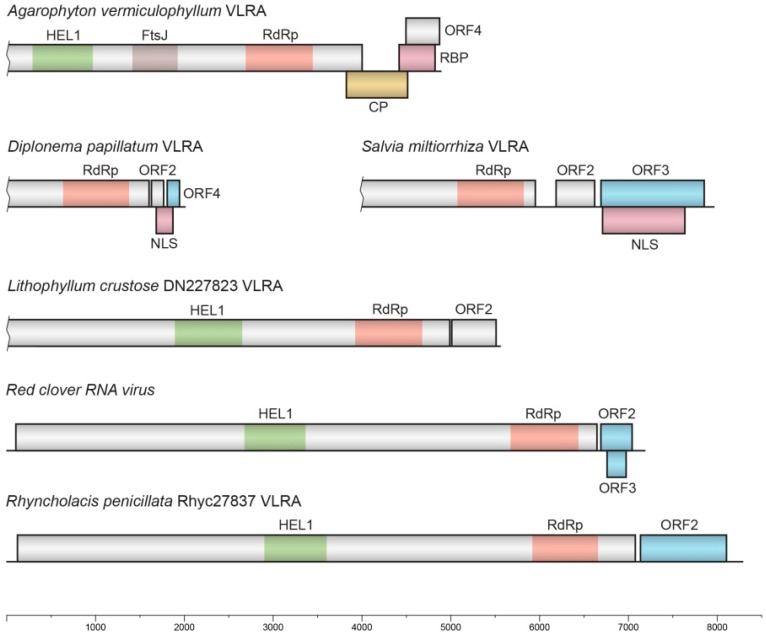
Gene organization of the selected plant viruses and VLRAs. The replicase protein domains from all VLRAs are shown (see text for details). CP—capsid protein; RBP—RNA-binding protein; NLS—protein with NLS signal. Proteins with putative membrane-bound segments are shown in blue.

**Figure 4 viruses-14-02680-f004:**
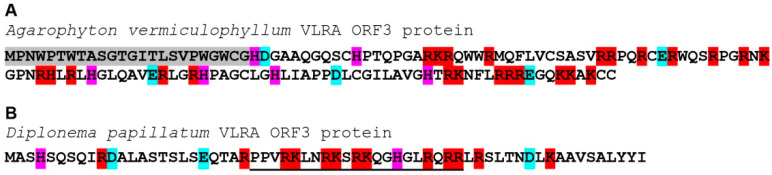
Amino acid sequence of the ORF3 protein encoded by Agarophyton vermiculophyllum (**A**) and Diplonema papillatum (**B**) VLRAs. Positively charged residues are shown in red, negatively charged residues are in blue, and histidine residues are in pink; uncharged N-terminal residues are shaded. The putative NLS sequence is underlined.

**Figure 5 viruses-14-02680-f005:**
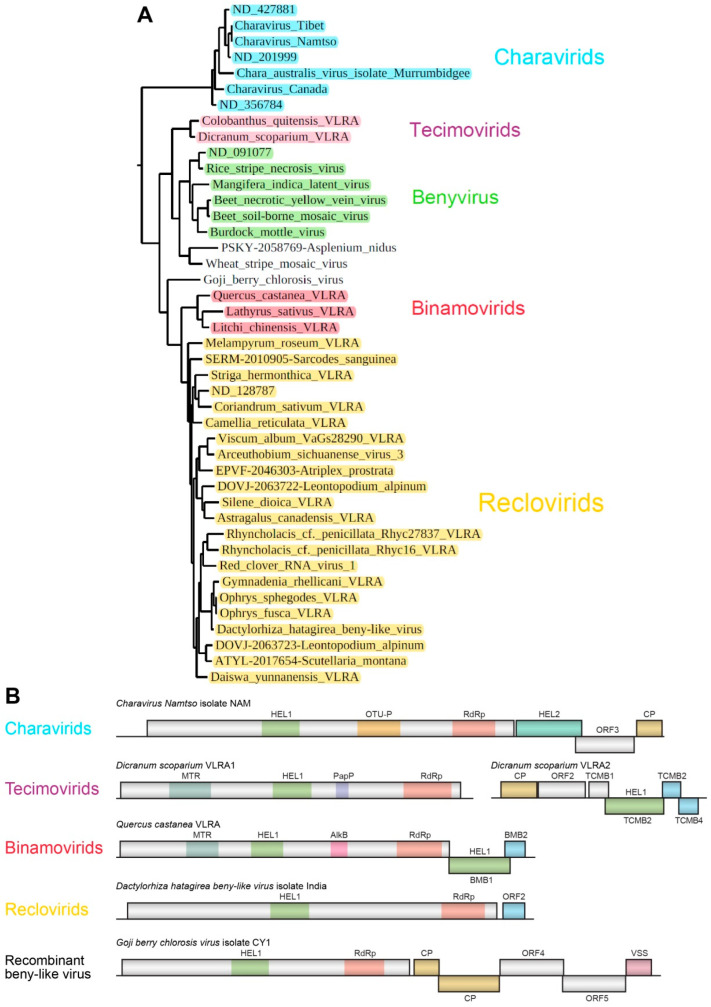
(**A**) Phylogenetic subtree of the whole tree of RdRp domains from Figure 2 showing a large cluster of Viridiplantae-specific viruses and derived from the aligned amino acid sequences of some benyviruses, benyvirus-like VLRAs and selected beny-like viruses (see text for details). (**B**) Gene organization of plant viruses and VLRAs that are representatives of Charavirids, Tecimovirids, Binamovirids and Reclovirids. The replicase protein domains from all VLRAs are shown (papain protease—PapP; OTU protease—OTU-P; other domains as in Figure 1). CP—capsid protein; VSS—virus silencing suppressor. Proteins with putative membrane-bound segments are shown in blue.

**Figure 6 viruses-14-02680-f006:**
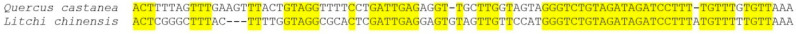
Pairwise nucleotide sequence alignment of the 3′-terminal regions preceding poly(A) in the predicted VLRA RNAs of binamovirids. Conserved nucleotides and RNA blocks are highlighted by yellow background. AAA at the 3′-terminus means poly(A) tract.

**Figure 7 viruses-14-02680-f007:**
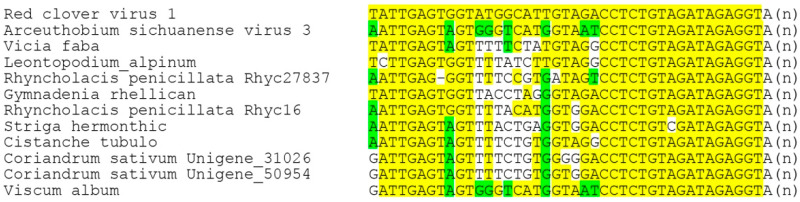
Multiple nucleotide sequence alignment of the 3′-terminal regions preceding poly(A) in the predicted VLRA RNAs of reclovirids. Conserved nucleotides and RNA blocks are highlighted by yellow background. Residues identical to *Arceuthobium sichuanense virus 3* are shown in green. A(n) at the 3′-terminus means poly(A) tract. Truncated VLRAs that are not shown in phylogenetic subtree include *Vicia faba* VLRA (GISP01006645) and *Cistanche tubulosa* VLRA (GJRS01079843).

**Figure 8 viruses-14-02680-f008:**
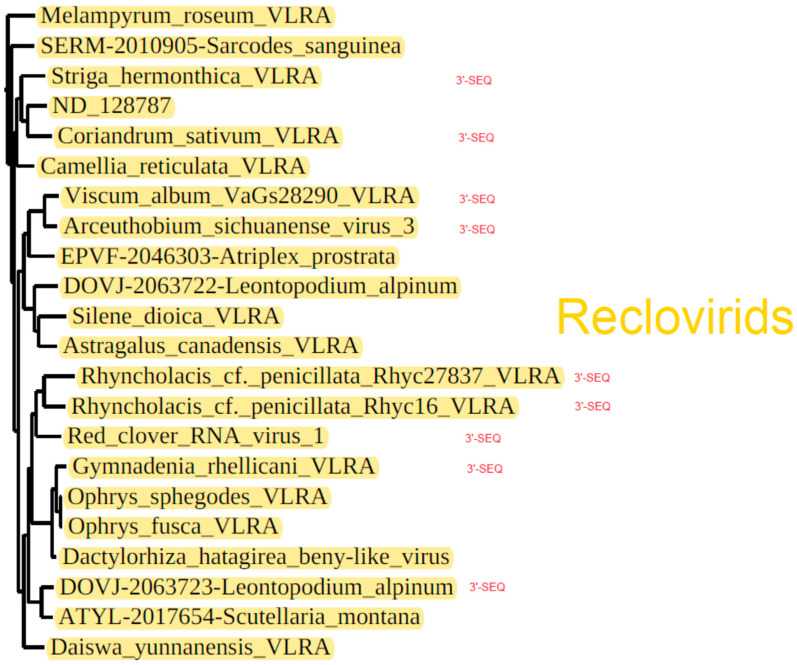
Phylogenetic subtree of the whole tree of RdRp domains from Figure 2 showing Reclovirid cluster of Viridiplantae-specific viruses. Viruses with conserved genomic 3′-termini are marked by 3′-SEQ (see text for details).

**Figure 9 viruses-14-02680-f009:**
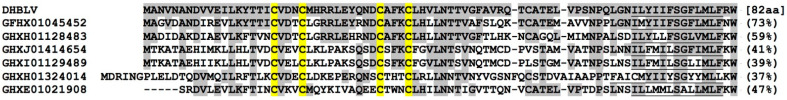
Multiple nucleotide sequence alignment of the N-terminal regions of ORF2 proteins encoded in the 3′-terminal regions of *Dactylorhiza hatagirea beny-like virus* (DHBLV) and predicted closely related VLRAs of reclovirids: *Dactylorhiza fuchsii* VLRA (GFHX01045452); *Gymnadenia rhellicani* VLRA TR40174 (GHXH01128483); *Ophrys sphegodes* VLRA (GHXJ01414654); *Ophrys fusca* VLRA (GHXI01129489); *Gymnadenia rhellicani* VLRA TR101479 (GHXH01324014); *Gymnadenia x densiflora* VLRA (GHXE01021908). Amino acid residues identical to those in DHBLV ORF2 protein are shaded. Conserved cysteines in the N-terminal sequences are in yellow. The hydrophobic sequences are underlined. The length of the presented DHBLV sequence is shown in square brackets, and identity to DHBLV protein is indicated on the right.

## Data Availability

All data are available upon reasonable request.

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
