# Peer review of "Uncovering Plant Virus Species Forming Novel Provisional Taxonomic Units Related to the Family *Benyviridae"

_viruses, 2022, doi:10.3390/v14122680_

Round 1

Reviewer 1 Report

In this manuscript, the authors analyzed molecular evolution of Benyvirus-Like RNAs and expanded the known host range of family Benyviridae to include red algae and charophyte algae based on the recent open-source data. They proposed several viruses as founding members of new potential taxons (sub- families or genera) in the virus family Benyviridae.

There are several issues needed to be addressed:

1.        Figure 1. The 5’-terminal “cap”-structure and a 3′ poly(A) tract must be included in the genomic organization of Beet necrotic yellow vein virus.

2.        Figure 2. please check the colors carefully. For example, yellow and blue represent arthropods and fungi, but Rhizoctonia solani and Bemisia tabaci were indicated by yellow and blue.   

3.        The plant-infecting benyviruses are usually multipartite or bipartite genomes. The authors showed several VLRAs were monopartite such as Chara australis virus. What is the evolutionary relatedness among the monopartite, bipartite, and multipartite genomes?

4.        It is better to evaluate the co-evolution of host and Benyvirus-Like RNAs in the results or discussion section.

5.        Line 275, “in Qinghai Lake in Tibet” should be “in Qinghai Lake in the Tibet Autonomous Region of China”

      6.      “3.3.4. Reclovirids” should be “3.3.5. Reclovirids” 

Author Response

We appreciate these helpful and encouraging reviewer’s comments very much.

  1. Figure 1. The 5’-terminal “cap”-structure and a 3′ poly(A) tract must be included in the genomic organization of Beet necrotic yellow vein virus.

The 5’-terminal “cap”-structure and a 3′ poly(A) tract are shown in Fig. 1.

  1. Figure 2. please check the colors carefully. For example, yellow and blue represent arthropods and fungi, but Rhizoctonia solani and Bemisia tabaci were indicated by yellow and blue.   

It was a mistake in figure legend. Now fungi are shown as yellow and arthropods - as blue. Bemisia tabaci is shown as arthropod.

3. The plant-infecting benyviruses are usually multipartite or bipartite genomes. The authors showed several VLRAs were monopartite such as Chara australis virus. What is the evolutionary relatedness among the monopartite, bipartite, and multipartite genomes?

The phrase “It should be noted that VLRAs of unknown origin (ND 427881, ND 201999 and ND 356784, see Figure 2 (https://riboviria.org/)) have genome organization resembling Chara australis virus” is added in the end of section 3.3.1. Additionally, the evolutionary relatedness among the monopartite, and multipartite beny-like-genomes is discussed throughout the text.

4. It is better to evaluate the co-evolution of host and Benyvirus-Like RNAs in the results or discussion section.

The co-evolution of host and plant virus represents important issue in plant molecular virology (see, for example, Richert-Pöggeler et al., Front Plant Sci. 2021, 12:689307. doi: 10.3389/fpls.2021.689307; Kulshrestha et al., Mol Biol Rep. 2020, 47(10):8219-8227. doi: 10.1007/s11033-020-05810-y). However, we believe that this issue is outside the scope of our paper. 

  1. Line 275, “in Qinghai Lake in Tibet” should be “in Qinghai Lake in the Tibet Autonomous Region of China”

It is changed accordingly (see section 3.3.1).

  1.     “3.3.4. Reclovirids” should be “3.3.5. Reclovirids” 

It is changed accordingly.

Reviewer 2 Report

Dear Authors,

In this study, the authors compared the partial genome sequences of some novel plus-RNA viruses from red algae, marine euglenozoan protist and land plants. Were reported sequence analyses of these genomes confirming previously reported evolutionary relationships between viruses of non-flowering plants, flowering plants, arthropods and fungi. The authors also suggest novel evolutionary links between viruses of red algae and land plants; and proposed viruses as founding members of new potential taxons (families or genera) related to family Benyviridae. The main novelty pointed out by the authors refers to identification of gene blocks coding for movement proteins in viruses of non-vascular plants indicating that beny- like viruses may have acquired movement proteins before to the emergence of the plant vascular system.

Based mainly on the RdRp domain, the authors bring relevant evolutionary aspects of the  beny-like viruses. RdRp is a importante hallmark protein and is commonly used in virus evolution studies.

It is also worth noting that only molecular data were used in the manuscript, including complete and partial sequences available in databases. The biological aspects indicated in the text have only a speculative aspect, except for the viruses that present characterization of biological aspects reported in the literature. Despite this, I consider that the data presented in the manuscript are relevant and bring news within the group of beny-like viruses.

Below are some points that should be corrected/included in the text:

1 - The first two paragraphs of the results should be inserted in the topic material and methods;

2 - Indicate in the methodology the average sizes of the analyzed sequences;

3 - I suggest using a larger number of replicates for the bootstrap percentage;

4 - The colors indicated in Figure 2 are not completely correct (check and correct);

5 - Check the writing of scientific names throughout the text, in several places the italic writing (correct form) was not applied (for example: lines 156, 172, 177 and 195). I emphasize that viral orders and families must also be written in italics. Viral species accepted by ICTV must also be written in italics, including in phylogenetic trees;

6 - The Wheat stripe mosaic virus and PSKY-2058769-Asplenium nidus were not classified in any genus and no observation and discussion was carried out regarding this.

Based on the above, I am in favor of publishing the article after minor revision.

Best regards

Author Response

We appreciate these helpful and encouraging reviewer’s comments very much.

1 - The first two paragraphs of the results should be inserted in the topic material and methods;

It is done (see para 2 and 4 of Material and methods).

2 - Indicate in the methodology the average sizes of the analyzed sequences;

It is done. “…the average sizes of the analyzed sequences were about 4500-5000 nucleotides” (see para 2 of Material and methods).

3 - I suggest using a larger number of replicates for the bootstrap percentage;

We used the number of replicates as recommended at http://www.phylogeny.fr/simple_phylogeny.cgi. The same number of replicates has been conventionally used in many previous papers (see, for example, Cabrera Mederos et al., Viruses. 2022, 14(10):2297. doi: 10.3390/v14102297; Shin et al., Viruses. 2022, 14(11):2376. doi: 10.3390/v14112376; Wang et al., Viruses. 2022,14(11):2369. doi: 10.3390/v14112369; Khalili et al., Viruses 2022,14(11):2325. doi: 10.3390/v14112325. Guo et al., Mitochondrial DNA B Resour. 2019, 4(2):3254-3255; Nam et al., Mitochondrial DNA B Resour. 2022, 7(1):167-169.

4 - The colors indicated in Figure 2 are not completely correct (check and correct);

It was a mistake in figure legend. It is now corrected as “fungi are shown as yellow and arthropods - as blue”.

5 - Check the writing of scientific names throughout the text, in several places the italic writing (correct form) was not applied (for example: lines 156, 172, 177 and 195). I emphasize that viral orders and families must also be written in italics. Viral species accepted by ICTV must also be written in italics, including in phylogenetic trees;

We have changed the name of viruses and taxons according to rules indicated in https://ictv.global/filebrowser/download/440.

6 - The Wheat stripe mosaic virus and PSKY-2058769-Asplenium nidus were not classified in any genus and no observation and discussion was carried out regarding this.

These species has been mentioned in section 3.3 (lines 2-6).

Reviewer 3 Report

The manuscript by Solovyev and Morozov described an in-silico analysis of plant viruses focusing on the family Benyviridae.

First, I will ask why this work is a communication article if it is clearly a research one.

In general, the manuscript is well written and the message that new viruses of this family were found is clear in the conclusion, but not in the abstract. I found the summary confusing as many ideas are drawn from a single phylogenetic tree with partial viral contigs as the source. The authors should keep the message simple and not exaggerate the findings.

The materials and methods section is too vague, there is a lack of explanation on how the flow of the results was obtained.

It is necessary to specify the raw number of analysed files, for example, 200 fastq files from 10 NCBI database bioprojects. Of these, a total of 50 viral contigs were found and used for further analysis. If not, it is unclear how comprehensive the analysis was.

A list of all data retrieved on the material and methods should be included. The IDs (ie, SRA fastq and BioProjects), the type of sequencing technology (short or long reads and SE or PE reads), and the data in Gb.

Also, the data availability section is empty, the viral contigs obtained here have not been deposited on any platform, such as NCBI Nucleotides with a GeneBank reference. This is essential for this article to be published.

In turn, the list of viral contigs with the corresponding GeneBank ids must be included in the material and methods and it must be specified in a column if the contig is a complete or partial sequence, in addition to the length of the contig.

Finally, I suggest changing the title to something more direct to the message.

“Uncovering novel virus and features from the family Benyviridae”.

Minor:

line 61 says “nucleotide and protein sequences” which is wrong, transcriptome databases only give you “nucleotides”, please change this.

Add the software used for the viral taxonomic classification as well as the one used to assess the quality of the contigs.

Author Response

We appreciate these helpful and encouraging reviewer’s comments very much.

1 - It is necessary to specify the raw number of analysed files, for example, 200 fastq files from 10 NCBI database bioprojects. Of these, a total of 50 viral contigs were found and used for further analysis. If not, it is unclear how comprehensive the analysis was.

A list of all data retrieved on the material and methods should be included. The IDs (ie, SRA fastq and BioProjects), the type of sequencing technology (short or long reads and SE or PE reads), and the data in Gb. 

First - Due to our text fault in Materials and methods, principal methodology has been misinterpreted by referees. We have stated in Materials and methods (para 1) that “To assemble the 3’-untranslated regions (up to 100-150 nucleotides) of some nearly full-length plant VLRAs, SRA experiments linked to the TSA projects were downloaded using fastq-dump tool of NCBI SRA Toolkit (http://ncbi.github.io/sra-tools/ accessed on 26 July 2022). Reads quality was checked with FastQC (https://www. bioinformatics. babraham.ac.uk/projects/fastqc/ accessed on 26 July 2022). De novo assembly of the 3’-untranslated regions of some VLRAs was carried out using SPAdes (http://cab.spbu.ru/software/spades/ accessed on 26 July 2022) [26] in “RNA mode”. In fact 98% of sequence information has been found in the NCBI TSA references (see section Materials and methods, para 3).

Second – according to the current NCBI rules sequences including significant sequence parts from existing TSA contigs are not accepted as third party sequences anymore.

2 - Also, the data availability section is empty, the viral contigs obtained here have not been deposited on any platform, such as NCBI Nucleotides with a GeneBank reference. This is essential for this article to be published.

All data are available upon reasonable request. See also answers 1 and 3.

3 - In turn, the list of viral contigs with the corresponding GeneBank ids must be included in the material and methods and it must be specified in a column if the contig is a complete or partial sequence, in addition to the length of the contig.

The list of viral contigs with the corresponding GeneBank ids is included in the material and methods (see Section 2, para 3).

4 - Finally, I suggest changing the title to something more direct to the message.

“Uncovering novel virus and features from the family Benyviridae”.

The title has been shortened to just “Uncovering virus species forming novel provisional taxonomic units related to the family Benyviridae”.

Minor:

5 - line 61 says “nucleotide and protein sequences” which is wrong, transcriptome databases only give you “nucleotides”, please change this.

It has been corrected.

6 - Add the software used for the viral taxonomic classification as well as the one used to assess the quality of the contigs.

Software indicated in Materials and Methods:  

The quality of the contigs -

https://www.ncbi.nlm.nih.gov/taxonomy/?term=viruses

Viral taxonomic classification -

https://www.bioinformatics. babraham.ac.uk/projects/fastqc/

Reviewer 4 Report

This paper reports the evolutional insight of benyvirus-like agents. The subject matter is interesting and suitable for publication in Viruses as a paper of Communications. However, I feel there are some issues that could be addressed before the acceptance of the manuscript. My comments are listed below. 

Comments:

Title: ~Benyvirus-Like RNA “Agents” in Algae?

Abstract:

-Regarding charophyte algae, the beny-like viruses have already been described by other research groups. 

-The “inter-order virus gene shuffling” should be explained more clearly in the main text.

1. Introduction: 

-The virus names described here (L25) and also hereafter (L39-43 and many others) may not be italicized (see ICTV site, https://ictv.global/faq/names). “Hepelivirales” should be in italic letter (L25).

-It would be nice if you could explain a little about other members of the order Hepelivirales. In addition, the information on classical benyviruses as plant pathogens may possibly be including this section.   

2. Materials and Methods

- Since it is not clear which VLRV sequences are newly discovered in this paper, it would be better to add a list including them. 

-The de novo assembled sequences derived from public transcriptome data should be deposited as TPA (Third Party Data) entries at GenBank. 

3. Results

-Classical benyviruses (BNYVV and BSBMV) are carry small RNA segments as part of their genome (Fig. 1). 

-In Fig 2, the color code is a bit difficult to distinguish (green and right green). Viruses from the charophite algae might be “light” blue in the legend (L146). “a closterovirus beet yellows virus“ as an outgroup (L141).

-No genes have been identified as coat protein (CP) for most of all beny-like viruses; therefore, the CP-like protein (CPL) is recommended for these CP candidates. 

- The subclade showing Fig. 2 can collapse here. Instead, please indicate the bootstrap values in Fig. 5. In addition, Fig. 8 might delete from the manuscript to aboide the data overlapping. The information about 3’-termini can also include in Fig. 5. 

-In Fig. 3, the representative benyvirus-like agents derived from insect and fungal samples or transcripts could be included here instead of plant viruses or VLRA, which can be presented later together with other plant benyvirus-like agents. 

-Some of the Figures, such as Figures 4, 6, and 9, may be moved as supplement items.

-For the five groups in the largest clade of the tree, please avoid adding "…virids" to each group name since it likely duplicates the family members such as benyvirids. Besides, the new item (a new figure), including representative genome structures of each proposed group, will be helpful to our understanding. It will be nice to clearly indicate the gene structures, which can potentially be responsible for cell-to-cell movement, for their discussion.

-The pairwise sequence comparison analysis using PASC or other bioinformatic tools can be included in the manuscript. This may help us to understand the relationships with and within each group of plant beny-like viruses.

4. Discussion

-Since this study is mainly data mining, it may be necessary to discuss concerns about miss-assembly, contamination, and the existence of unknown segments.

-The authors could be include the discussion of the plant benyvirid taxonomy together with the species demarcation criteria set for classical benyviruses.

-The CP-readthrough domains are involved in the vector transmission of classical benyviruses. Therefore, the presence or absence of similar gene units in the plant beny-like viruses should be worthy of descriving and also discussion.

Author Response

 We appreciate these helpful and encouraging reviewer’s comments very much.

  1. Title: ~Benyvirus-Like RNA “Agents” in Algae?

Regarding charophyte algae, the beny-like viruses have already been described by other research groups. 

The “inter-order virus gene shuffling” should be explained more clearly in the main text.

Title and the corresponding places in Abstract have been changed.

  1. The virus names described here (L25) and also hereafter (L39-43 and many others) may not be italicized (see ICTV site, https://ictv.global/faq/names). “Hepelivirales” should be in italic letter (L25).

We have changed the names of viruses and taxons according to https://ictv.global/filebrowser/download/440.

  1. It would be nice if you could explain a little about other members of the order Hepelivirales. In addition, the information on classical benyviruses as plant pathogens may possibly be including this section.

The information on other families of order Hepelivirales has been added (see Section 2, para 2, last sentence).     

  1. Since it is not clear which VLRV sequences are newly discovered in this paper, it would be better to add a list including them. 

Corresponding list of VLRAs has been added to Section 2, para 3.

  1. The de novo assembled sequences derived from public transcriptome data should be deposited as TPA (Third Party Data) entries at GenBank. 

First - Due to our text fault in Materials and methods, principal methodology has been misinterpreted by referees. Now, we have stated in Materials and methods (para 1) that “To assemble the 3’-untranslated regions (up to 100-150 nucleotides) of some nearly full-length plant VLRAs, SRA experiments linked to the TSA projects were downloaded using fastq-dump tool of NCBI SRA Toolkit (http://ncbi.github.io/sra-tools/ accessed on 26 July 2022). Reads quality was checked with FastQC (https://www. bioinformatics. babraham.ac.uk/projects/fastqc/ accessed on 26 July 2022). De novo assembly of the 3’-untranslated regions of some VLRAs was carried out using SPAdes (http://cab.spbu.ru/software/spades/ accessed on 26 July 2022) [26] in “RNA mode”. In fact 98% of sequence information has been found in the NCBI TSA references (see section Materials and methods, para 3).

Second – according to the current NCBI rules sequences including significant sequence parts from existing TSA contigs are not accepted as third party sequences anymore.

  1. Classical benyviruses (BNYVV and BSBMV) are carry small RNA segments as part of their genome (Fig. 1). 

It is now corrected (see Introduction, para 1).

  1. In Fig 2, the color code is a bit difficult to distinguish (green and right green). Viruses from the charophite algae might be “light” blue in the legend (L146). “a closterovirus beet yellows virus“as an outgroup (L141).

It is done (Fig.2 legend).

  1. No genes have been identified as coat protein (CP) for most of all beny-like viruses; therefore, the CP-like protein (CPL) is recommended for these CP candidates. 

It has been corrected accordingly (see Section 3.1, para 4; Section 3.3.2, para 1; Section 3.3.3, para 1; Discussion, para 1).

  1. The subclade showing Fig. 2 can collapse here. Instead, please indicate the bootstrap values in Fig. 5. In addition, Fig. 8 might delete from the manuscript to aboide the data overlapping. The information about 3’-termini can also include in Fig. 5. 

We believe that the current structure of illustrations reflects the main text in a best way.

  1. In Fig. 3, the representative benyvirus-like agents derived from insect and fungal samples or transcripts could be included here instead of plant viruses or VLRA, which can be presented later together with other plant benyvirus-like agents. 

This manuscript describes plant benyvirus-like agents mainly and only transiently insect and fungal viruses.

  1. Some of the Figures, such as Figures 4, 6, and 9, may be moved as supplement items.

We believe that these figures are important for fruitful reading the main text.

  1. For the five groups in the largest clade of the tree, please avoid adding "…virids" to each group name since it likely duplicates the family members such as benyvirids. Besides, the new item (a new figure), including representative genome structures of each proposed group, will be helpful to our understanding. It will be nice to clearly indicate the gene structures, which can potentially be responsible for cell-to-cell movement, for their discussion.

Accordingly, we have included a new figure (Fig. 5B) to show representative genome structures of novel beny-like virus groups.

  1. The pairwise sequence comparison analysis using PASC or other bioinformatic tools can be included in the manuscript. This may help us to understand the relationships with and within each group of plant beny-like viruses.

In relation to these questions, please see: First, Materials and Methods and Supplementary tables 1-3; Second, pairwise sequence comparisons are now discussed in the text (Section 3.1, para 2;  para 5; Section 3.2, para 3; Section 3.3.5, para 5).

  1. Since this study is mainly data mining, it may be necessary to discuss concerns about miss-assembly, contamination, and the existence of unknown segments.

Now these concerns are mentioned in Conclusion.

  1. The authors could be include the discussion of the plant benyvirid taxonomy together with the species demarcation criteria set for classical benyviruses.

We have included some discussions related to taxon’s demarcation among novel potential beny-like viruses. See, Section 3.1, para 2; para 5; Section 3.2, para 3; Section 3.3.5, para 5.

  1. The CP-readthrough domains are involved in the vector transmission of classical benyviruses. Therefore, the presence or absence of similar gene units in the plant beny-like viruses should be worthy of describing and also discussion.

In general, this manuscript contains no descriptions of beny-like viruses with CP-readthrough domains.